# Safe-Sora: Safe Text-to-Video Generation via Graphical Watermarking

**Zihan Su**[1]    **Xuerui Qiu**[2,3]    **Hongbin Xu**[4]    **Tangyu Jiang**[1]    **Junhao Zhuang**[1]
**Chun Yuan**[1*]   **Ming Li**[5*]   **Shengfeng He**[6]    **Fei Richard Yu**[5]

[1] Tsinghua Shenzhen International Graduate School, Tsinghua University
[2] Institute of Automation, Chinese Academy of Sciences
[3] Zhongguancun Academy [4] Bytedance
[5] Guangdong Laboratory of Artificial Intelligence and Digital Economy (SZ)
[6] Singapore Management University
zh-su24@mails.tsinghua.edu.cn

## Abstract

The explosive growth of generative video models has amplified the demand for reliable copyright preservation of AI-generated content. Despite its popularity in image synthesis, invisible generative watermarking remains largely underexplored in video generation. To address this gap, we propose *Safe-Sora*, the first framework to embed graphical watermarks directly into the video generation process. Motivated by the observation that watermarking performance is closely tied to the visual similarity between the watermark and cover content, we introduce a hierarchical *coarse-to-fine adaptive matching mechanism*. Specifically, the watermark image is divided into patches, each assigned to the most visually similar video frame, and further localized to the optimal spatial region for seamless embedding. To enable spatiotemporal fusion of watermark patches across video frames, we develop a 3D wavelet transform-enhanced Mamba architecture with a novel *spatiotemporal local scanning strategy*, effectively modeling long-range dependencies during watermark embedding and retrieval. To the best of our knowledge, this is the first attempt to apply state space models to watermarking, opening new avenues for efficient and robust watermark protection. Extensive experiments demonstrate that *Safe-Sora* achieves state-of-the-art performance in terms of video quality, watermark fidelity, and robustness, which is largely attributed to our proposals. Code is publicly available at https://github.com/Sugewud/Safe-Sora

## 1 Introduction

Recent advances in video generation models have significantly transformed digital content creation [1–6]. VideoCrafter2 [2] delivers high-fidelity video generation results, while Open-Sora [7] enables efficient and scalable video generation. However, this rapid progress also raises growing concerns over copyright protection and ownership verification of generated videos.

Invisible watermarking has proven effective for copyright protection in image generation [8–15]. However, its extension to video generation remains relatively underexplored. Recent efforts such as VideoShield [16] and LVMark [17] embed watermarks by modifying latent noise or applying importance-based modulation strategies. Despite these advancements, existing approaches rely on embedding bitstring-based identifiers, which fall short of leveraging the high information capacity inherent in video content. Unlike static images, videos offer significantly greater embedding bandwidth, making them well-suited for graphical watermarks—e.g., logos or icons—that serve as more

---

*Corresponding authors.

39th Conference on Neural Information Processing Systems (NeurIPS 2025).

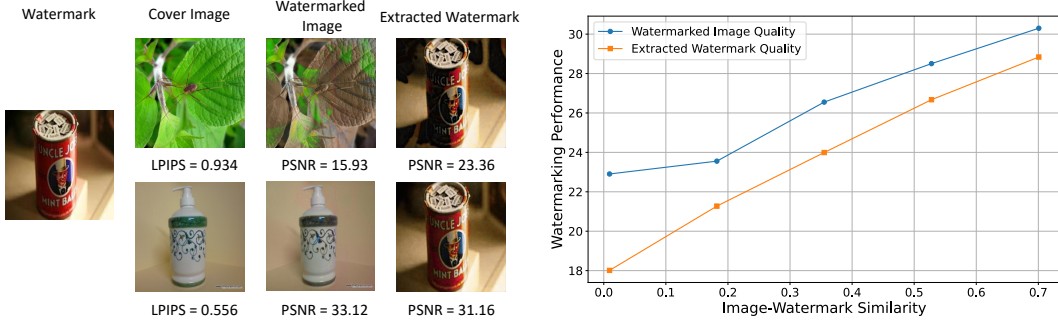

Figure 1: Impact of image-watermark similarity on watermarking performance. We used a pretrained classic image hiding network Balujanet [18] on 1,000 image pairs, each consisting of a graphical watermark from Logo-2k [19] and a cover image from ImageNet [20]. Image-Watermark similarity was quantified using 1-LPIPS and the quality of the watermarked image and extracted watermark was evaluated using PSNR. Higher PSNR and lower LPIPS indicate improved performance.

intuitive and visually recognizable evidence of ownership. Such designs enhance both the perceptual clarity and practical reliability of copyright verification.

Recognizing the untapped potential of graphical watermarking in video generation, we propose Safe-Sora, the first framework, to the best of our knowledge, that embeds graphical watermarks elegantly into the video generation process. As illustrated in Fig. 1, we observe that watermarking performance significantly correlates with the visual similarity between the watermark and cover images. In particular, embedding becomes significantly more effective when the cover image shares high visual similarity with the watermark content. Motivated by this, we propose a hierarchical coarse-to-fine adaptive matching mechanism, which first divides the watermark image into patches and assigns each patch to the most similar video frame through an *inter-frame* automatic selection strategy. Subsequently, an *intra-frame* localization is performed to embed the patch into the most visually similar region within the selected frame. To address the challenge of fusing and extracting watermark information distributed across spatiotemporal locations, we further propose a 3D wavelet transform-enhanced Mamba architecture with a tailored scanning strategy. This design enables bidirectional modeling across frequency subbands in the 3D wavelet transform, effectively and efficiently capturing long-range dependencies in both space and time. To the best of our knowledge, this is the first application of state space models to generative watermarking.

In our experiments, we utilize the widely-used Panda-70M [21] dataset as the video source due to its extensive scale and diverse video categories. For graphical watermarks, we employ the Logo-2K+ [19] dataset, which offers a wide variety of real-world logos. The quantitative and qualitative comparisons with existing methods demonstrate that the proposed Safe-Sora achieves state-of-the-art performance in terms of video quality, watermark fidelity, and robustness. For instance, our method achieves a Fréchet Video Distance of 3.77, far lower than the second-best baseline's 154.35, highlighting its superior temporal consistency. Our primary contributions can be summarized as follows:

- We introduce the first model specifically designed to embed graphical watermarks in video generation pipelines, directly addressing the pressing need for copyright protection of generated video content.
- We propose a hierarchical coarse-to-fine adaptive matching mechanism that strategically embeds watermark patches into visually similar frames and spatial regions, enhancing overall watermarking performance.
- We pioneer the application of state space models for watermarking through a novel 3D wavelet transform-enhanced Mamba architecture with a tailored scanning strategy, enabling enhanced fusion and extraction of watermark information across space and time.

## 2 Related Work

### 2.1 Video Diffusion Models

Recently, AI-generated content has been vibrant in the community [22–33]. Diffusion models [34–39] are a class of generative models that synthesize data through a gradual denoising process, beginning

from randomly sampled Gaussian noise. Latent Video Diffusion Models (LVDMs) [40] perform the diffusion process in the latent space to improve computational efficiency. VideoCrafter2 [2] builds high-quality video generation models by leveraging low-quality video data combined with synthesized high-quality images. Open-Sora [7] introduces the Spatial-Temporal Diffusion Transformer, an efficient video diffusion framework that separates spatial and temporal attention mechanisms. While LVDMs have shown strong performance in video generation, the integration of graphical watermarks into this framework has not been explored.

## 2.2 Generative Video Watermarking

Digital watermarking has emerged as an essential technique for copyright protection, content authentication, and ownership verification across various media types. However, watermarking for video diffusion models represents a relatively unexplored area. VideoShield [16] pioneered this space by modifying latent noise during the diffusion process to embed binary watermark information. More recently, LVMark [17] introduced an importance-based weight modulation strategy to minimize visual quality degradation. Nevertheless, these existing approaches primarily focus on embedding low-capacity binary strings, without taking advantage of the high-capacity nature of video media, which is well-suited for embedding richer information such as graphical watermarks.

## 2.3 State Space Models

State Space Models (SSMs) [41, 42] have emerged as efficient alternatives to transformers [43] for sequence modeling. The Mamba architecture [44] represents a significant advancement in SSMs by introducing selective state space modeling with data-dependent parameters, enabling dynamic resource allocation to important sequence elements while maintaining computational efficiency. Despite Mamba's remarkable success in language processing tasks [45, 46] and its growing adoption in computer vision applications [47, 48], its potential for watermarking techniques has remained entirely unexplored until now.

# 3 Graphical Watermarking for Video Generation

In this section, we present the pipeline of our Safe-Sora framework, which introduces a novel approach to embedding graphical watermarks directly within the video generation process (Fig. 2). We first partition the watermark image into patches and optimally assign them to appropriate video frames and regions (Section 3.1). These patches are then embedded and upsampled to generate the watermark feature map. To embed the watermark, this feature map is fused with multi-scale video features using a UNet built with 2D SFMamba blocks (Section 3.2), followed by a series of 3D SFMamba blocks that leverage our spatiotemporal local scanning strategy (Section 3.3), producing a watermarked video. To extract the watermark, the watermarked video is processed through an extraction network built with a degradation layer, a series of 3D SFMamba blocks, and position recovery. The training objectives are outlined in Section 3.4, while the preliminaries on latent video diffusion models, state space models, and wavelet transforms are detailed in Appendix A.

## 3.1 Coarse-to-Fine Adaptive Patch Matching

Motivated by the observation that greater similarity between the watermark and cover content enhances watermarking performance (as shown in Fig. 1), we propose a *coarse-to-fine adaptive patch matching* mechanism to systematically identify the most semantically similar spatial-temporal regions in a video for watermark embedding, as illustrated in the bottom-left corner of Fig. 2.

First, to enable accurate localization of each patch during the final watermark recovery, we propose a simple yet effective method: the position channel. Specifically, we represent patch positions using binary encoding (e.g., using 8 bits to represent 256 patch positions). This binary code is then replicated to form an additional channel, introducing redundancy that enhances robustness against spatial distortions and degradation. Finally, this position channel is concatenated with the patch content, embedding positional information directly into the input and eliminating the need for additional positional processing during subsequent training.

Then, we adopt a two-stage process to adaptively determine the most suitable embedding location for each patch. The first stage operates at the frame level. We extract features from both patches and

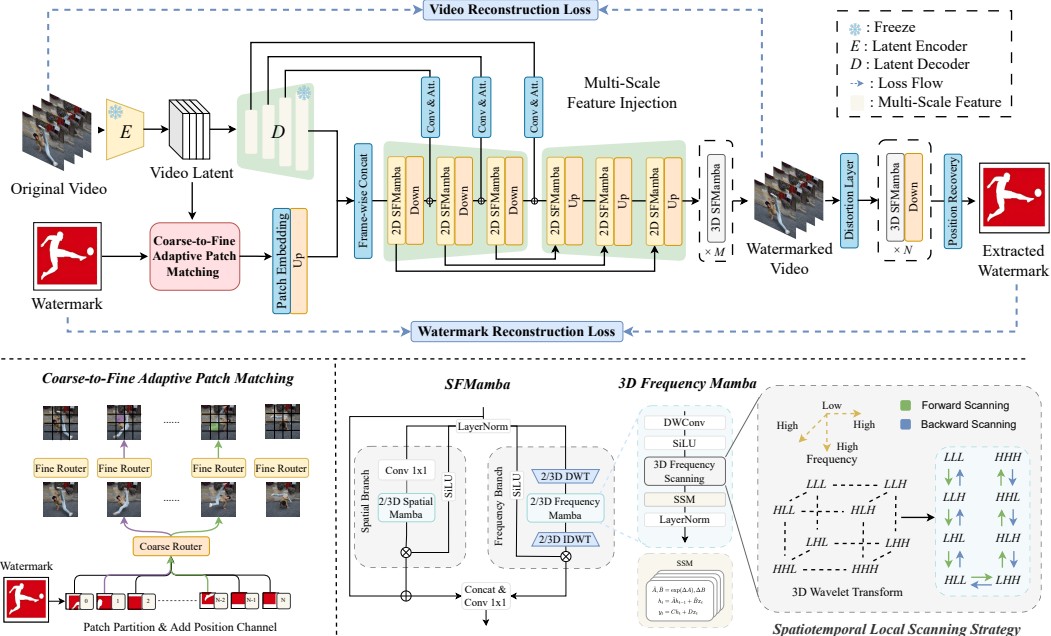

Figure 2: Overview of our Safe-Sora framework. Our method consists of three main components: (1) Coarse-to-Fine Adaptive Patch Matching: partitioning the watermark image into patches and optimally assigning them to appropriate video frames and regions, followed by patch embedding and upsampling to generate the watermark feature map; (2) Watermark Embedding: the watermark feature map is fused with multi-scale video features via a UNet with 2D SFMamba blocks, followed by a series of 3D SFMamba blocks that implement our spatiotemporal local scanning strategy, to produce the watermarked video; (3) Watermark Extraction: recovering the embedded watermark using an extraction network built with a distortion layer, a series of 3D SFMamba blocks, and position recovery. The difference between different types of Mamba blocks lies in their scanning strategies.

the latent representations of video frames using a convolution layer followed by ReLU and global average pooling (GAP). Similarity between each patch $i$ and frame $j$ is computed via dot product of these feature vectors, and normalized using Softmax:

$$\mathbf{w}_{i,j} = \text{Softmax}\left(\text{GAP}(\text{ReLU}(\text{Conv}(\mathbf{p}_i))) \cdot \text{GAP}(\text{ReLU}(\text{Conv}(\mathbf{z}_j)))\right). \quad (1)$$

Here, $\mathbf{w}_{i,j}$ denotes the similarity score between patch $\mathbf{p}_i$ and the latent representation $\mathbf{z}_j$ of frame $j$. Each patch is then assigned to the frame with the highest similarity score. To ensure balanced distribution, we impose a maximum capacity for each frame. If the top-ranked frame is full, the patch is redirected to the next highest available candidate. Having selected a frame, we proceed to the fine stage, which determines the optimal spatial position within that frame. Each frame is subdivided into spatial regions according to its patch capacity. Feature representations of these regions are computed similarly, and the similarity between patch $i$ and region $k$ in the assigned frame $j$ is given by:

$$\mathbf{s}_{i,k} = \text{Softmax}\left(\text{GAP}(\text{ReLU}(\text{Conv}(\mathbf{p}_i))) \cdot \text{GAP}(\text{ReLU}(\text{Conv}(\mathbf{r}_{j,k})))\right), \quad (2)$$

where $\mathbf{s}_{i,k}$ is the similarity score between the $i$-th patch and the $k$-th region $\mathbf{r}_{j,k}$ in the latent representation of frame $j$. Note that we take full advantage of the inherent feature properties of latent variables in video generation models. Since latent variables can already be viewed as feature extractions of the original frames, we use only a single convolutional layer for feature extraction, which significantly reduces the computational overhead.

## 3.2 Spatial-Frequency Mamba for Spatial Fusion

Mamba [44] has demonstrated strong capabilities in modeling long-range dependencies with high efficiency, making it well-suited for spatiotemporal modeling in video tasks. Meanwhile, frequency domain information has been applied in various domains [49–51]. In watermark embedding, it has proven effective in capturing structural patterns and resisting distortions [50, 51]. To incorporate both advantages, we propose the Spatial-Frequency Mamba (SFMamba) block, as shown in Fig. 2.

SFMamba adopts a dual-stream design with separate spatial and frequency branches. It comes in two variants: a 2D version and a 3D version, differing primarily in the wavelet transform and scanning

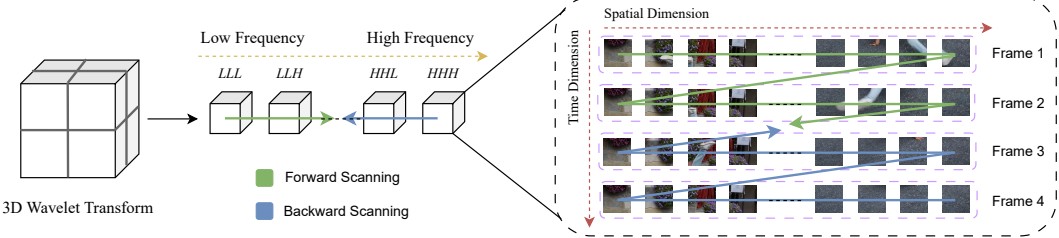

Figure 3: For 3D frequency scanning, we propose a spatiotemporal local scanning strategy for 3D wavelet transform, which processes the frequency components hierarchically from low frequency to high frequency and high frequency to low frequency.

strategy. The 3D SFMamba will be introduced in Section 3.3. We next introduce the 2D SFMamba block for efficient spatial fusion of watermark and video content. It consists of separate 2D spatial and 2D frequency branches.

**2D Spatial Branch.** The spatial processing begins with a LayerNorm operation on the input feature map $\mathbf{F}_{\text{in}}$, yielding normalized features $\mathbf{F}_{\text{N}}$. In the first path, $\mathbf{F}_{\text{N}}$ undergoes a simple SiLU activation function. In the second path, $\mathbf{F}_{\text{N}}$ passes through a $1\times1$ convolution layer, followed by our 2D spatial Mamba module. The 2D spatial branch output $\mathbf{F}_s$ is computed as:

$$\mathbf{F}_s = \text{SiLU}(\mathbf{F}_{\text{N}}) \odot 2\text{DSpatialMamba}(\text{Conv}_{1\times1}(\mathbf{F}_{\text{N}})). \tag{3}$$

where $\odot$ denotes element-wise multiplication of the two pathway outputs.

**2D Frequency Branch.** For frequency domain processing, we transform $\mathbf{F}_{\text{N}}$ using a 2D Discrete Wavelet Transform (DWT), which decomposes the signal into four frequency subbands: *LL* (low-low), *LH* (low-high), *HL* (high-low), and *HH* (high-high). Each subband has spatial dimensions reduced by half compared to the original. Inspired by FreqMamba [52], we rearrange these components from top-left to bottom-right to restore the original resolution. The wavelet features are then divided into four blocks and scanned block by block. The output is projected back to the spatial domain via a 2D Inverse DWT (IDWT), followed by element-wise multiplication with $\text{SiLU}(\mathbf{F}_{\text{N}})$. The 2D frequency branch output $\mathbf{F}_f$ is computed as:

$$\mathbf{F}_f = \text{SiLU}(\mathbf{F}_{\text{N}}) \odot \text{IDWT}(2\text{DFreqMamba}(\text{DWT}(\mathbf{F}_{\text{N}}))). \tag{4}$$

The spatial branch output is enhanced with a residual connection from $\mathbf{F}_{\text{in}}$. Finally, we concatenate the outputs from both branches and apply a $1\times1$ convolution to produce the integrated output.

### 3.3 3D Frequency Scanning for Spatiotemporal Interaction

To address the challenges of fusing and extracting watermark information distributed across spatiotemporal locations, we propose an efficient architecture—3D SFMamba, a 3D Wavelet Mamba transform-enhanced design with a customized scanning strategy. This architecture enables bidirectional modeling across frequency subbands within the 3D wavelet transform, effectively capturing long-range dependencies in both spatial and temporal domains to accurately recover watermark information embedded in the temporal dimension. 3D SFMamba consists of separate 3D spatial and 3D frequency branches.

**3D Spatial Branch.** The 3D spatial branch employs a vanilla 3D scanning strategy, which processes features across all three dimensions (temporal, height, width) to capture both spatial and temporal dependencies effectively.

**3D Frequency Branch.** In the frequency domain branch, input features $\mathbf{F}_{\text{in}}$ undergo a 3D Discrete Wavelet Transform (3D DWT), decomposing them into eight subbands: *LLL*, *LLH*, *LHL*, *LHH*, *HLL*, *HLH*, *HHL*, and *HHH*. Each subband has half the original dimensions in frame, height, and width. To address the complexity of 3D wavelet-transformed features, we propose a novel spatiotemporal local scanning strategy as shown in Fig. 3. This approach first rearranges the eight subbands to restore the original video resolution, then divides them into eight distinct parts for separate scanning. For forward scanning, the order follows *LLL*, *LLH*, *LHL*, *HLL*, *LHH*, *HLH*, *HHL*, and *HHH*—progressing systematically from low to high frequencies. Additionally, we implement a reverse scanning mechanism that processes the subbands in the opposite direction—from *HHH* to *LLL*—enabling the model to capture information from high to low frequencies. Within each part, we employ a spatial-first, temporal-second scanning pattern. This spatiotemporal local scanning

strategy is specifically designed for 3D wavelet transforms, allowing the model to process frequency information hierarchically across multiple scales.

## 3.4 Training Objectives

Our training framework combines video reconstruction loss and watermark reconstruction loss. The video reconstruction loss uses mean squared error (MSE) to ensure the watermarked video $\hat{\mathbf{V}}$ closely resembles the original video $\mathbf{V}$:

$$\mathcal{L}_{\text{video}} = \text{MSE}(\mathbf{V}, \hat{\mathbf{V}}). \tag{5}$$

Similarly, the watermark reconstruction loss measures the extraction accuracy by comparing the extracted watermark $\hat{\mathbf{W}}$ with the original watermark $\mathbf{W}$:

$$\mathcal{L}_{\text{watermark}} = \text{MSE}(\mathbf{W}, \hat{\mathbf{W}}). \tag{6}$$

During training, we provide the correct positions to reconstruct the watermark image properly, while during testing, the model utilizes the embedded position channels to predict the correct arrangement of patches. The final loss function is:

$$\mathcal{L}_{\text{total}} = \mathcal{L}_{\text{video}} + \lambda \, \mathcal{L}_{\text{watermark}}, \tag{7}$$

where the watermark weighting hyperparameter $\lambda$ balances video quality against watermark fidelity.

# 4 Experiments

## 4.1 Experimental Setting

**Datasets.** For the video dataset, we use the Panda-70M [21] dataset for training, which is a large-scale dataset containing 70 million high-quality videos across diverse content types. Specifically, we randomly download 10,000 videos from Panda-70M, sample 8 frames from each video, and resize each frame to a resolution of $320 \times 512$ for training purposes. For the watermark dataset, we use the Logo-2K dataset [19], which contains 167,140 watermark images at a resolution of $256 \times 256$, spanning a wide range of real-world logo classes. For the evaluation of text-to-video generation, we employ the VidProm [53] dataset as the source of prompts. The prompts in VidProm are generated by GPT-4 [54], and we randomly select 100 prompts from the dataset for evaluation.

**Implementation Details.** We use VideoCrafter2 [2] as our backbone model to generate videos at a resolution of $320 \times 512$. Our method is compatible with various video generation backbones, with additional results provided in Appendix C. The patch size is set to $16 \times 16$. Patch Embedding maps each patch to a 1024-dimensional feature space. The model is trained for 30 epochs on 4 NVIDIA RTX 4090 GPUs. We adopt the AdamW optimizer [55], with the initial learning rate set to 5e-4, which is gradually decayed to 1e-6 following a cosine decay schedule. The watermark embedding network uses $M = 2$ 3D SFMamba Blocks, while the watermark extraction network uses $N = 4$ 3D SFMamba Blocks. The hyperparameter $\lambda$ in Eq. 7 is set to 0.75. The distortion layer simulates various real-world distortions, including H.264 video compression, rotation, and other common transformations. Since H.264 is non-differentiable, we follow DVMark [56] and use a 3D CNN to mimic its effects. For position recovery, we propose a confidence-guided greedy assignment algorithm, with detailed descriptions provided in Appendix B.

**Baselines.** To the best of our knowledge, no existing method embeds graphical watermarks directly into video generation models. To provide a comprehensive comparison, we select five representative state-of-the-art methods spanning three distinct paradigms of graphical watermarking: (1) Post-processed image watermarking methods: **Balujanet**[18] – A classic image steganography network; **UDH**[57] – A classic graphical watermarking network; **PUSNet** [58] – A state-of-the-art image steganography network. (2) Generative image watermarking: **Safe-SD** [59] – A generative graphical watermarking approach. (3) Video steganography: **Wengnet** [60] – A method that hides one video within another. For a fair comparison, we retrain all baseline methods using the same training dataset as ours. For image-based methods, we embed a complete watermark image into each frame. For video-based methods, each frame of the secret video acts as a watermark and is embedded into the corresponding frame of the cover video.

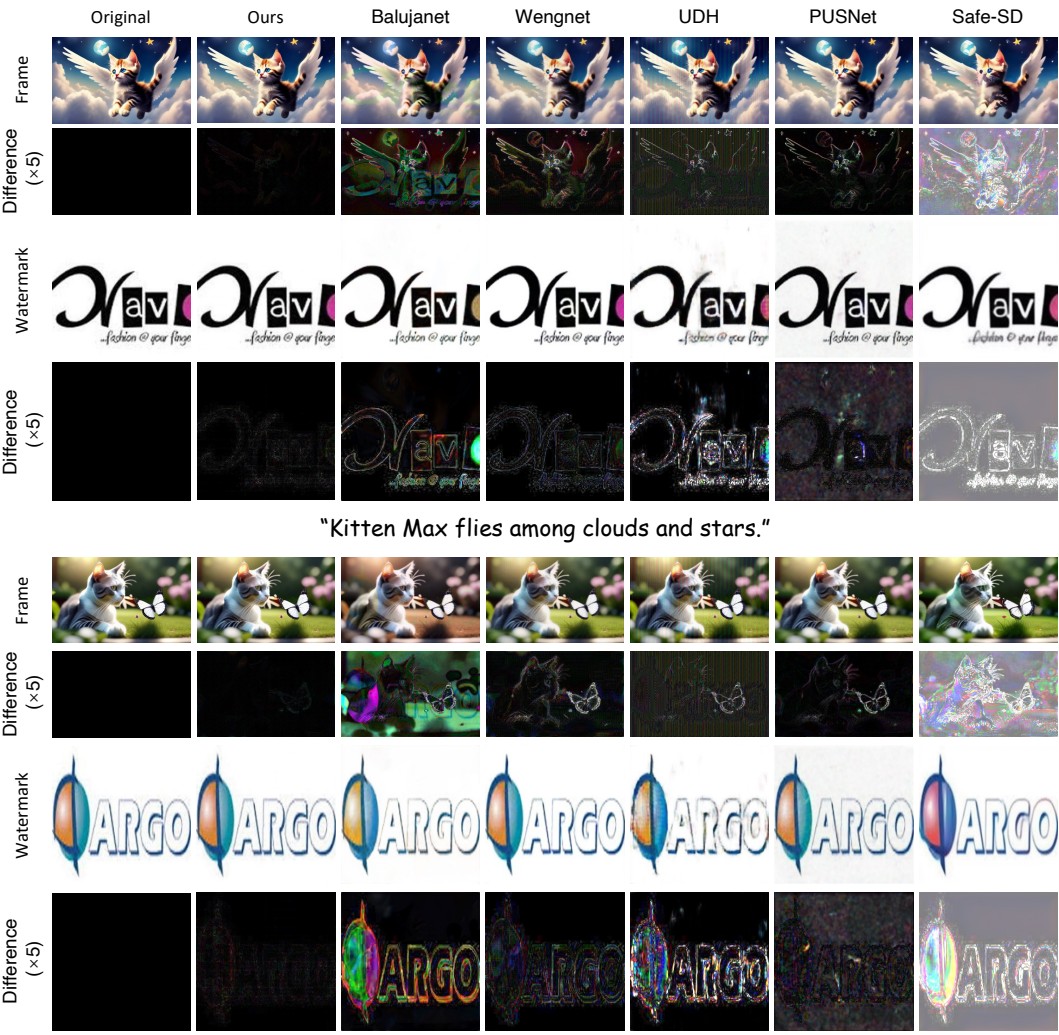

|  | Original | Ours | Balujanet | Wengnet | UDH | PUSNet | Safe-SD |

"Kitten Max flies among clouds and stars."

"A gray and white cat is playing with a butterfly in a beautiful garden."

Figure 4: Qualitative comparison results on the first frame of each video. Difference maps show absolute differences between the watermarked and original videos, and between the recovered and original watermarks. More examples are shown in Fig. 10 of Appendix. **Best viewed with zoom in.**

## 4.2 Comparison with State-of-the-art Methods

**Qualitative Comparison.** Fig. 4 shows the qualitative comparisons on the first frame of each video, while Fig. 5 presents visual results of Safe-Sora across multiple frames. As illustrated, Balujanet introduces clearly visible artifacts in the watermarked video, UDH suffers from stripe-like distortions, and Safe-SD presents noticeable color shifts. From the difference maps, it is evident that both WengNet and PUSNet introduce considerable degradation to both video quality and watermark fidelity. In contrast, our method produces watermarked videos with high visual fidelity, exhibiting minimal differences from the original videos. Moreover, the recovered watermark images closely resemble the originals, demonstrating high reconstruction accuracy.

**Quantitative Comparison.** To evaluate the accuracy of watermark recovery and the invisibility of the watermark (i.e., video quality), we adopt standard metrics including PSNR, MAE, RMSE, SSIM [61], and LPIPS [62]. To assess temporal consistency in videos, we employ tLP [63] and Fréchet Video Distance (FVD) [64]. Quantitative results are summarized in Tab 1. As shown in the table, our method achieves state-of-the-art performance across all evaluation metrics. We observe that image watermarking methods inject watermarks by embedding them independently into each frame, which leads to poor temporal consistency and higher FVD scores. In contrast, our method leverages Mamba's long-range modeling capability across space and time, along with the proposed

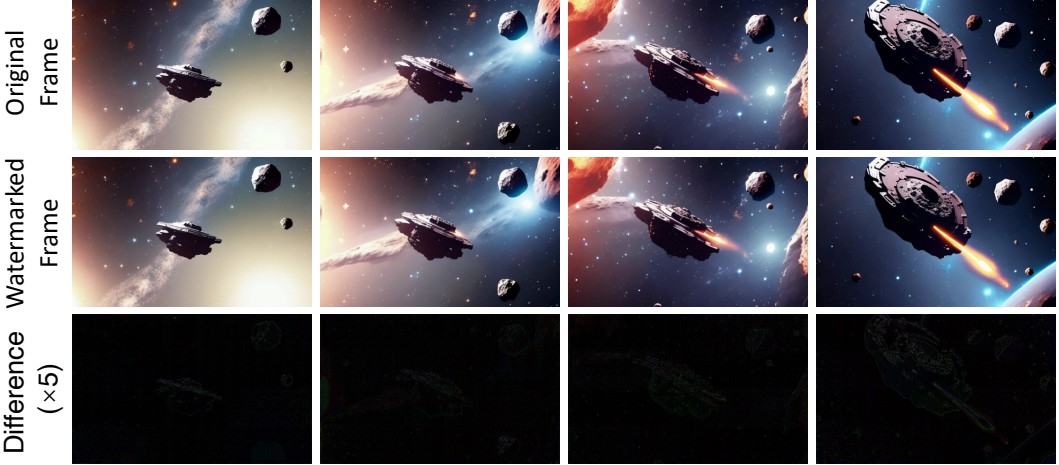

"Spaceships traverse a vibrant cosmos filled with planets and stars."

Figure 5: Visual results of Safe-Sora on multiple frames. For each frame, we show the original image, the corresponding watermarked image, and their residual difference. **Best viewed with zoom in.**

Table 1: Quantitative results on watermark quality and video quality metrics. Watermark quality is measured by comparing the recovered watermark image with the original watermark, while video quality is evaluated by comparing the watermarked video with the original video.

| Method | Watermark quality | | | | | Video quality | | | | | | |
| --- | --- | --- | --- | --- | --- | --- | --- | --- | --- | --- | --- | --- |
| | PSNR ↑ | MAE ↓ | RMSE ↓ | SSIM ↑ | LPIPS ↓ | PSNR ↑ | MAE ↓ | RMSE ↓ | SSIM ↑ | LPIPS ↓ | tLP ↓ | FVD ↓ |
| Balujanet | 25.28 | 9.61 | 15.10 | 0.91 | 0.11 | 25.26 | 10.09 | 14.58 | 0.87 | 0.25 | 1.32 | 512.22 |
| Wengnet | 33.18 | 3.71 | 5.82 | 0.96 | 0.06 | 28.09 | 6.27 | 10.69 | 0.85 | 0.21 | 1.27 | 265.82 |
| UDH | 22.90 | 11.29 | 19.29 | 0.77 | 0.24 | 27.75 | 8.16 | 10.72 | 0.73 | 0.32 | 2.09 | 1075.62 |
| PUSNet | 28.86 | 7.45 | 9.57 | 0.93 | 0.12 | 29.98 | 4.50 | 8.72 | 0.92 | 0.11 | 0.98 | 154.35 |
| Safe-SD | 24.24 | 9.78 | 17.39 | 0.84 | 0.11 | 22.32 | 11.65 | 20.64 | 0.75 | 0.24 | 1.87 | 849.83 |
| Ours | **37.71** | **2.22** | **3.61** | **0.97** | **0.04** | **42.50** | **1.36** | **1.96** | **0.98** | **0.01** | **0.38** | **3.77** |

spatiotemporal local scanning strategy, resulting in superior temporal consistency. Specifically, our method achieves an FVD of 3.77, significantly outperforming all baselines.

## 4.3 Robustness

To rigorously evaluate the robustness of our method, we apply a variety of distortion types. For random erasing, we randomly select an erasure ratio from the range [5%, 10%, 15%, 20%]. For Gaussian blur, we randomly choose a kernel size from 3, 5, 7. For Gaussian noise, we add noise with a standard deviation randomly sampled from a uniform distribution $\mathcal{U}(0, 0.2)$. For rotation, the degree is randomly sampled from the range $(-30°, 30°)$. Specifically for video, we adopt H.264 compression with a fixed CRF value of 24. We use PSNR, SSIM, and LPIPS to evaluate the robustness of watermark reconstruction under these distortions. As shown in Fig. 6, our method consistently achieves the best performance across all types of attacks, demonstrating strong robustness. In particular, under H.264 compression, all baseline methods suffer a significant drop in performance, whereas our method maintains high watermark quality.

## 4.4 Ablation Study

We conduct an ablation study on two key components— Coarse-to-Fine Adaptive Patch Matching and Spatiotemporal Local Scanning. Additional ablation studies can be found in Appendix D.

**Impact of Coarse-to-Fine Adaptive Patch Matching.** This strategy matches the most similar frame and spatial location for each watermark patch, based on similarity computed with the video latent representations. To evaluate the effectiveness of each component, we investigate three ablated variants of our method: **w/o CFAPM**, which completely removes the Coarse-to-Fine Adaptive Patch Matching mechanism; **w/o RtL**, which replaces the Routing by Latent strategy with a direct pixel-frame similarity computation; and **w/o FS**, which removes the Fine Stage responsible for spatial location refinement.

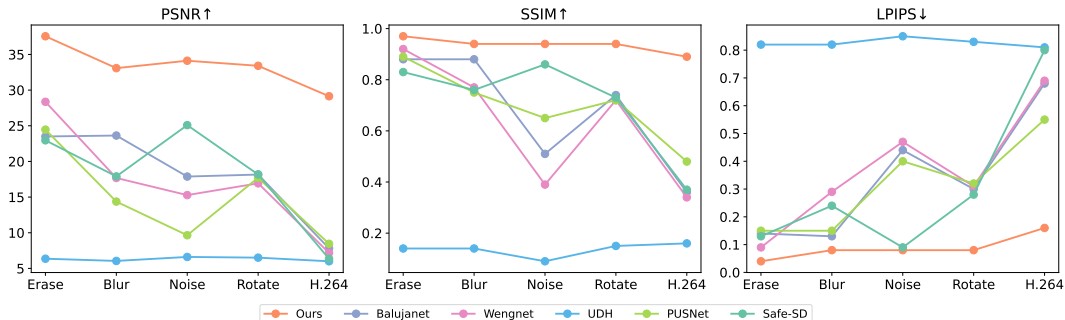

Figure 6: Watermark reconstruction quality under various distortions. Distortion settings include: Random Erasing (5%–20%), Gaussian Blur (kernel size 3/5/7), Gaussian Noise ($\sigma \sim \mathcal{U}(0, 0.2)$), Rotation (-30°, 30°), and H.264 Compression (CRF = 24).

Table 2: Comprehensive ablation study on key components of our method. CFAPM: Coarse-to-Fine Adaptive Patch Matching; RtL: Routing by Latent; FS: Fine Stage; SLS: Spatiotemporal Local Scanning; SFS: Spatial First Scanning within each subband.

| Method | Watermark quality | | | | | Video quality | | | | | | |
|---|---|---|---|---|---|---|---|---|---|---|---|---|
| | PSNR ↑ | MAE ↓ | RMSE ↓ | SSIM ↑ | LPIPS ↓ | PSNR ↑ | MAE ↓ | RMSE ↓ | SSIM ↑ | LPIPS ↓ | tLP ↓ | FVD ↓ |
| w/o CFAPM | 36.71 | 2.53 | 3.99 | 0.96 | 0.05 | 39.68 | 1.94 | 2.76 | 0.97 | 0.03 | 1.14 | 16.87 |
| w/o RtL | 36.36 | 2.67 | 4.13 | 0.96 | 0.05 | 40.23 | 1.79 | 2.54 | 0.97 | 0.04 | 1.30 | 6.37 |
| w/o FS | 36.88 | 2.45 | 3.94 | **0.97** | **0.04** | 41.25 | 1.58 | 2.26 | 0.97 | 0.03 | 1.17 | 4.82 |
| w/o SLS | 35.96 | 2.98 | 4.02 | 0.94 | 0.08 | 38.42 | 1.98 | 2.12 | 0.92 | 0.03 | 1.01 | 13.16 |
| w/o SFS | 36.41 | 2.59 | 4.17 | 0.96 | 0.05 | 42.21 | 1.38 | 2.05 | **0.98** | **0.01** | **0.24** | 5.24 |
| Ours | **37.71** | **2.22** | **3.61** | **0.97** | **0.04** | **42.50** | **1.36** | **1.96** | **0.98** | **0.01** | 0.38 | **3.77** |

The results in Tab. 2 clearly demonstrate that each component of the CFAPM strategy plays a critical role in enhancing overall performance. Computing the similarity between watermark patches and video latents leverages the compressed semantic information encoded in the latent space, enabling more accurate matching; the fine stage further refines this process by identifying the most visually similar spatial location for each patch. Overall, the Coarse-to-Fine Adaptive Patch Matching mechanism consistently improves both watermark fidelity and video quality.

**Impact of Spatiotemporal Local Scanning.** This strategy traverses the eight subbands of the 3D wavelet transform in a frequency-aware hierarchical order. Within each subband, patches are selected following a spatial-first, temporal-second scanning pattern. To evaluate the effectiveness of this design, we ablate two key components: **w/o SLS**, which replaces the structured traversal with a vanilla 3D scanning strategy; and **w/o SFS**, which applies a temporal-first scanning order within each subband instead of the proposed spatial-first policy.

Results in Tab. 2 demonstrate that the full SLS strategy significantly improves both watermark and video quality. While the temporal-first scanning achieves slightly better tLP, it consistently underperforms in watermark fidelity metrics. In summary, SLS enables more effective fusion and extraction of watermark signals distributed across spatiotemporal regions, thereby enhancing the overall performance of watermark embedding.

## 5 Conclusion

Our work introduces Safe-Sora, the first framework embedding graphical watermarks directly into generated video. We propose a hierarchical coarse-to-fine adaptive matching strategy that optimally maps watermark patches to visually similar frames and spatial regions. Our 3D wavelet transform-enhanced Mamba architecture with a novel spatiotemporal local scanning strategy, effectively models spatiotemporal dependencies for watermark embedding and retrieval, pioneering the application of state space models to watermarking. Experiments demonstrate that Safe-Sora achieves superior performance in video quality, watermark fidelity, and robustness. This work establishes a foundation for copyright protection in generative video and opens new avenues for applying state space models to digital watermarking.

## Acknowledgments

This work is supported by the National Key R&D Program of China (2022YFB4701400/4701402), SSTIC Grant (KJZD20230923115106012, KJZD20230923114916032, GJHZ20240218113604008), and the National Natural Science Foundation of China (62502317).

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

# Technical Appendices

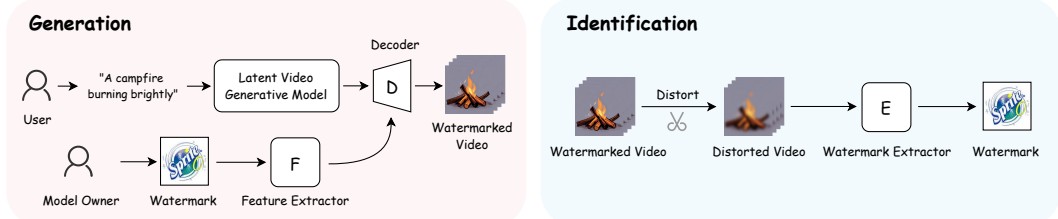

Figure 7: Application Scenario of Safe-Sora: A user provides a text prompt to a video generation model. The model owner's graphical watermark is embedded into the video through a feature extractor and decoder. Later, even if the video is distorted, a watermark extractor can recover the graphical watermark to verify authenticity and ensure copyright protection.

## A  Preliminaries

### A.1  Latent Video Diffusion Models

Latent Video Diffusion Models (LVDMs) extend the concept of latent diffusion models to the video domain. These models operate in a compressed latent space rather than pixel space to improve computational efficiency while maintaining generation quality. The process can be described in three key steps:

First, a video encoder $\mathcal{E}$ maps the input video $x \in \mathbb{R}^{F \times H \times W \times 3}$ to a latent representation $z = \mathcal{E}(x) \in \mathbb{R}^{F \times h \times w \times c}$, where $F$ is the number of frames, and the spatial dimensions are reduced: $h < H$ and $w < W$.

Second, a diffusion process gradually adds noise to the latent representation through a fixed Markov chain:

$$q(z_t|z_{t-1}) = \mathcal{N}(z_t; \sqrt{1-\beta_t}z_{t-1}, \beta_t \mathbf{I}), \tag{8}$$

$$q(z_t|z_0) = \mathcal{N}(z_t; \sqrt{\bar{\alpha}_t}z_0, (1-\bar{\alpha}_t)\mathbf{I}), \tag{9}$$

where $\beta_t$ is the noise schedule, $\alpha_t = 1 - \beta_t$, and $\bar{\alpha}_t = \prod_{s=1}^{t} \alpha_s$.

Finally, a denoising network $\epsilon_\theta$ is trained to predict the added noise at each time step. During generation, the reverse process starts from pure Gaussian noise $z_T \sim \mathcal{N}(0, \mathbf{I})$ and iteratively denoises to produce $z_0$, which is then decoded to the final video $\hat{x} = \mathcal{D}(z_0)$ using a decoder $\mathcal{D}$.

For text-to-video generation, LVDMs incorporate a text encoder that processes a conditioning prompt, which guides the denoising process toward the desired content.

### A.2  State Space Models

State Space Models (SSMs) are continuous dynamical systems defined by the following equations:

$$\frac{d\mathbf{h}(t)}{dt} = \mathbf{A}\mathbf{h}(t) + \mathbf{B}\mathbf{x}(t), \tag{10}$$

$$\mathbf{y}(t) = \mathbf{C}\mathbf{h}(t) + \mathbf{D}\mathbf{x}(t), \tag{11}$$

where $\mathbf{x}(t)$ is the input, $\mathbf{h}(t)$ is the hidden state, $\mathbf{y}(t)$ is the output, and $\{\mathbf{A}, \mathbf{B}, \mathbf{C}, \mathbf{D}\}$ are the parameters of the system.

For discrete sequence modeling, these continuous equations are discretized:

$$\mathbf{h}_t = \bar{\mathbf{A}}\mathbf{h}_{t-1} + \bar{\mathbf{B}}\mathbf{x}_t, \tag{12}$$

$$\mathbf{y}_t = \mathbf{C}\mathbf{h}_t + \mathbf{D}\mathbf{x}_t, \tag{13}$$

where $\bar{\mathbf{A}}$ and $\bar{\mathbf{B}}$ are the discretized versions of $\mathbf{A}$ and $\mathbf{B}$.

---
**Algorithm 1** Confidence-Guided Greedy Assignment for Watermark Position Recovery

---
1: **Input:** Watermark patches with position channel
2: **Output:** Reconstructed watermark image $\mathcal{W}$

---
    *Stage 1: Position Decoding*
3: **for** each patch $i$ **do**
4:     Normalize position channel to $[0, 1]$
5:     Compute probability vector $p_i$ by averaging binary vectors in the position channel
6:     Compute confidence $c_i = \frac{1}{K} \sum_{j=1}^{K} |p_i^j - 0.5|$
7:     Convert $p_i$ to binary $\hat{b}_i$ via thresholding
8:     Decode $\hat{b}_i \rightarrow$ position index $pos_i \in [0, N-1]$
9: **end for**

---
    *Stage 2: Confidence-Prioritized Assignment*
10: Initialize watermark image $\mathcal{W} \leftarrow \emptyset$
11: Initialize unassigned patch pool $\mathcal{U} \leftarrow \emptyset$
12: **for** each patch $i$ **do**
13:     **if** $pos_i$ is unoccupied in $\mathcal{W}$ **then**
14:         Assign patch $i$ to position $pos_i$ in $\mathcal{W}$
15:     **else if** $c_i >$ confidence of current patch at $pos_i$ **then**
16:         Replace patch at $pos_i$ with $i$ in $\mathcal{W}$
17:         Add the replaced patch to $\mathcal{U}$
18:     **else**
19:         Add patch $i$ to $\mathcal{U}$
20:     **end if**
21: **end for**

---
    *Stage 3: Greedy Reassignment of Unassigned Patches*
22: Sort $\mathcal{U}$ by descending $c_i$
23: **for** each patch $j$ in $\mathcal{U}$ **do**
24:     Find nearest vacant position $p_j$ to $pos_j$
25:     Assign patch $j$ to position $p_j$ in $\mathcal{W}$
26: **end for**
27: **return** $\mathcal{W}$

---

The Mamba architecture extends traditional SSMs by introducing input-dependent parameters:

$$\bar{\mathbf{A}}, \bar{\mathbf{B}} = \text{Projection}(\mathbf{x}), \tag{14}$$

$$\mathbf{h}_t = \bar{\mathbf{A}} \odot \mathbf{h}_{t-1} + \bar{\mathbf{B}} \odot \mathbf{x}_t, \tag{15}$$

$$\mathbf{y}_t = \mathbf{C}\mathbf{h}_t, \tag{16}$$

This input-dependent parameterization allows Mamba to dynamically adapt its processing based on input content, making it effective for modeling complex sequential dependencies.

## A.3   Wavelet Transforms

Wavelet transforms decompose signals into multiple frequency components with localized time information, making them useful for frequency domain watermarking.

For images, the 2D Discrete Wavelet Transform (DWT) decomposes an image into four sub-bands: approximation (*LL*), horizontal detail (*LH*), vertical detail (*HL*), and diagonal detail (*HH*).

The 3D Discrete Wavelet Transform extends the 2D DWT to the temporal domain for video processing. A video sequence is decomposed into eight sub-bands: *LLL*, *LLH*, *LHL*, *LHH*, *HLL*, *HLH*, *HHL*, and *HHH*, with *L* and *H* representing low and high frequencies across the frame, height, and width dimensions. Each sub-band has half the resolution of the original video in all dimensions. The 3D DWT provides a multi-level representation of videos, capturing both spatial and temporal characteristics, which is beneficial for video watermarking by allowing embedding in specific frequency bands while preserving perceptual quality.

Table 3: Quantitative comparison on VideoCrafter2 and Open-Sora backbones.

| Backbone | Watermark quality | | | | | Video quality | | | | | | |
|---|---|---|---|---|---|---|---|---|---|---|---|---|
| | PSNR ↑ | MAE ↓ | RMSE ↓ | SSIM ↑ | LPIPS ↓ | PSNR ↑ | MAE ↓ | RMSE ↓ | SSIM ↑ | LPIPS ↓ | tLP ↓ | FVD ↓ |
| VideoCrafter2 | **37.71** | **2.22** | **3.61** | **0.97** | **0.04** | 42.50 | 1.36 | 1.96 | **0.98** | **0.01** | 0.38 | 3.77 |
| Open-Sora | 35.42 | 2.93 | 4.70 | 0.96 | 0.06 | **44.15** | **1.31** | **1.75** | 0.97 | **0.01** | **0.31** | **3.04** |

| Frame | Watermaked frame | Difference (×5) | Watermark | Recovered watermark | Difference (×5) |
|---|---|---|---|---|---|

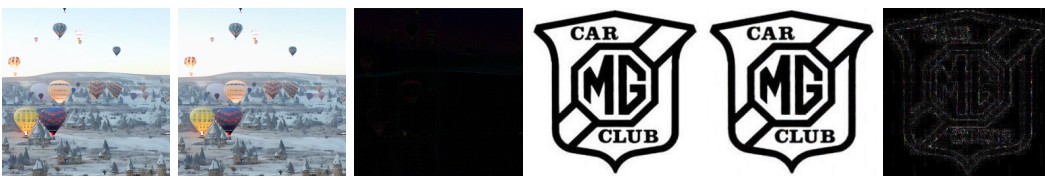

"A flock of seagulls flies over the azure sea and above the red cliffs."

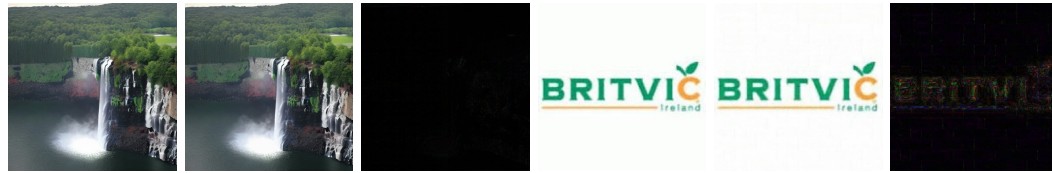

"Numerous hot air balloons float above a snow-covered, peculiar landscape."

"A magnificent waterfall cascades amidst the lush forest."

Figure 8: Qualitative examples on Open-Sora backbone. Best viewed with zoom in.

## B   Robust Watermark Position Recovery Algorithm

To address rare cases where multiple watermark patches are decoded to the same spatial location due to distortion or attack, we propose a confidence-guided greedy assignment algorithm. This algorithm ensures reliable and unambiguous recovery of watermark positions by incorporating confidence estimation, conflict resolution, and greedy reassignment of unplaced patches.

The algorithm is as follows: first, compute the confidence score for each patch's predicted position. Then, assign each patch to its corresponding position; in case of conflicts, give priority to the patch with higher confidence. Finally, assign the remaining unplaced patches in descending order of confidence to the nearest available positions. The detailed procedure is illustrated in Algorithm 1.

The confidence-guided greedy assignment algorithm effectively handles noisy or partial position corruption and significantly improves the robustness of watermark extraction.

## C   More Backbones

While our main experiments are conducted using VideoCrafter2 [2], a UNet-based video generation model, we further evaluate our method using Open-Sora [7], a DiT-based video generation model. Quantitative results are shown in Tab. 3, and qualitative examples are provided in Fig. 8. As can be seen, Open-Sora achieves comparable performance to VideoCrafter2 and produces videos with higher visual quality, but slightly lower watermark fidelity. These results demonstrate that our method is effective across different video generation models.

Table 4: Additional Ablation Studies. MSFI: Multi-Scale Feature Injection.

| Method | Watermark quality | | | | | Video quality | | | | | | |
|---|---|---|---|---|---|---|---|---|---|---|---|---|
| | PSNR ↑ | MAE ↓ | RMSE ↓ | SSIM ↑ | LPIPS ↓ | PSNR ↑ | MAE ↓ | RMSE ↓ | SSIM ↑ | LPIPS ↓ | tLP ↓ | FVD ↓ |
| w/o MSFI | 36.56 | 2.56 | 4.06 | 0.96 | 0.05 | 39.39 | 2.02 | 2.84 | 0.97 | 0.03 | 1.19 | 14.11 |
| Ours | **37.71** | **2.22** | **3.61** | **0.97** | **0.04** | **42.50** | **1.36** | **1.96** | **0.98** | **0.01** | **0.38** | **3.77** |

| Original | w/o MSFI | w/ MSFI | | Original | w/o MSFI | w/ MSFI |

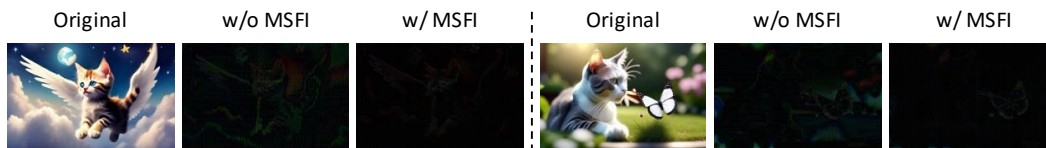

Figure 9: Visual impact of Multi-Scale Feature Injection. We present difference maps (×5) between watermarked and original videos. After applying Multi-Scale Feature Injection, the differences are significantly reduced, leading to improved video quality.

## D  Additional Ablation Studies

To further assess the contribution of individual components in our framework, we perform extended ablation studies beyond the main experiments. In particular, we examine the impact of Multi-Scale Feature Injection, with quantitative results reported in Tab. 4 and qualitative comparisons shown in Fig. 9. The results demonstrate that incorporating the inherent multi-scale features of the VAE notably improves the visual quality of generated videos.

## E  Limitations

While our method demonstrates strong performance in embedding and recovering static graphical watermarks, it is currently limited to image-based watermarks such as logos or icons. Embedding more complex and information-rich video watermarks—e.g., animated sequences or temporally dynamic patterns—remains a challenge.

## F  Societal Impact

The ability to embed graphical watermarks directly into the video generation process carries important social and ethical implications. On the positive side, it provides a practical solution to the growing concerns over ownership verification and copyright protection in generative media. As synthetic content becomes increasingly widespread, methods like ours can help content creators assert their rights and trace misuse, thereby fostering accountability and transparency in digital media ecosystems.

However, like many watermarking techniques, our method may also be misused. For example, it could potentially be employed to falsely claim ownership over public material, or to embed unauthorized logos into generated videos. We strongly advocate for the responsible use of generative watermarking technologies and recommend that future research explores methods to verify the authenticity of embedded watermarks and prevent abuse.

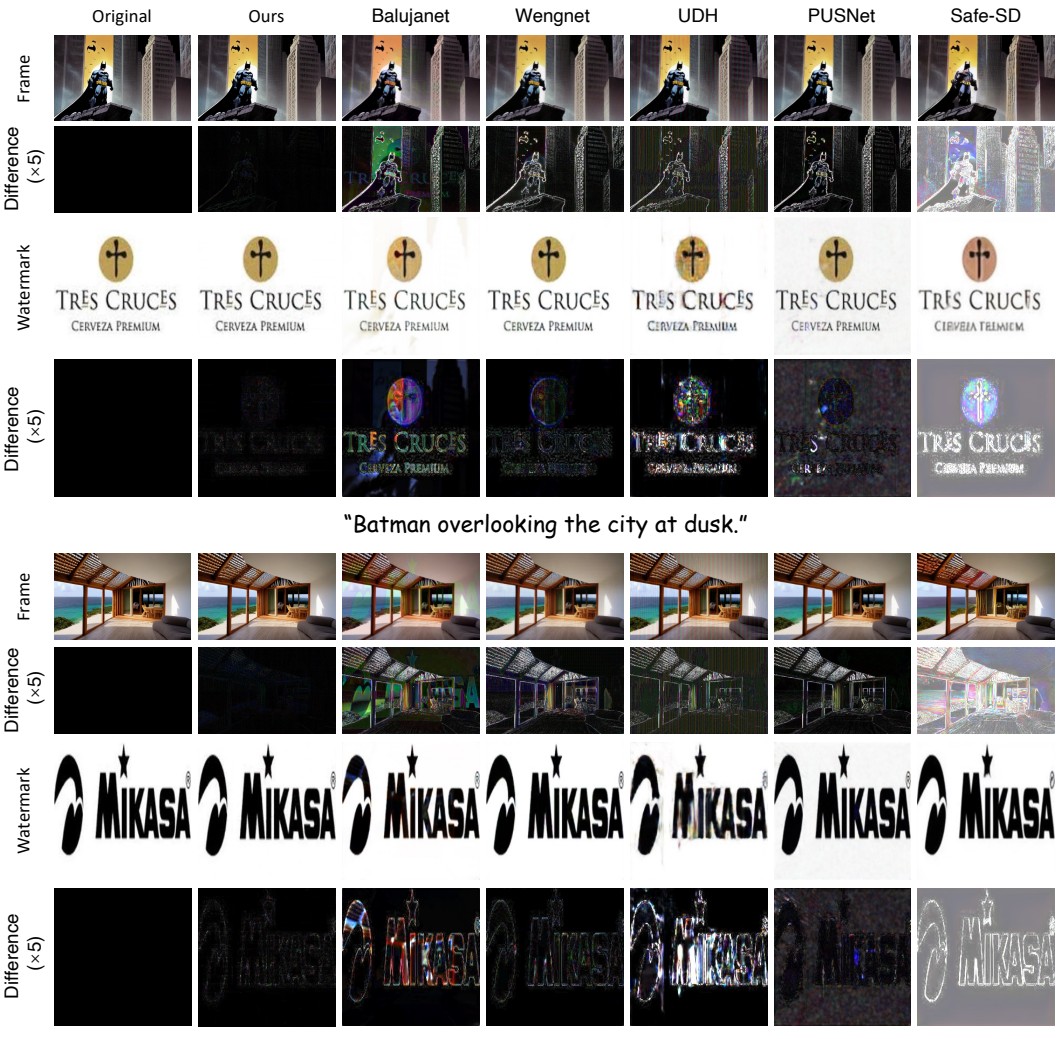

Figure 10: More qualitative examples on VideoCrafter2 backbone. Difference maps show absolute differences between the watermarked and original videos, and between the recovered and original watermarks. **Best viewed with zoom in.**

