# OpenReview forum: "Safe-Sora: Safe Text-to-Video Generation via Graphical Watermarking"
_NeurIPS.cc/2025/Conference — NeurIPS 2025 poster_

### Official Review · Reviewer_VQjW · 2025-06-24

**Clarity:** 3
**Significance:** 2
**Originality:** 3
**Rating:** 5
**Confidence:** 4

**Summary:**

This paper presents Safe-Sora, a novel framework that embeds graphical watermark into the video generation process. To minimize the impact of watermark embedding on video quality, the authors propose a coarse-to-fine adaptive matching mechanism. In addition, a state space model is employed to fuse watermark and video features, effectively preserving the spatiotemporal consistency of the generated video. Experimental results demonstrate that the proposed method can embed image watermarks while maintaining high visual quality for both the video and the watermark, and exhibits strong robustness against various distortions.

**Questions:**

1. In response to Weakness #1, it is recommended to clearly elaborate on how the proposed method differs from post-processing watermarking methods.
2. In response to Weakness #2, a detailed analysis of the time or computational complexity should be provided.
3. In the performance comparison experiments (Section 4.2), it is advisable to include comparisons with methods specifically designed for image steganography, such as HiNet[2], and to replace unsuitable baselines PUSNet.
4. In the robustness experiments (Section 4.3), additional tests should be conducted under distortions such as random crop, frame insert, frame drop, and frame swap. Relevant experimental setups can refer to VideoShield[3].
5. In Section D of the Technical Appendices: Additional Ablation Studies, while the impact of Multi-Scale Feature Injection is explored, the strategy used for feature injection after removing this component is not mentioned. This should be clarified.
6. In the right subfigure of Fig. 1, it is suggested to revise the y-axis label from "watermarking performance" to PSNR, which would provide a clearer and more precise representation.
7. Last but not least, given that the watermark embedding occurs at the feature in the decoder, a critical question arises: if a watermarked video is first encoded back into latent space using the same encoder, and then decoded again, would this process effectively remove the watermark? This scenario should be considered and discussed in the paper.

[2] Jing, J., Deng, X., Xu, M., Wang, J., & Guan, Z. (2021). Hinet: Deep image hiding by invertible network. In Proceedings of the IEEE/CVF international conference on computer vision (pp. 4733-4742).
[3] Hu, R., Zhang, J., Li, Y., Li, J., Guo, Q., Qiu, H., & Zhang, T. (2025). VideoShield: Regulating Diffusion-based Video Generation Models via Watermarking. arXiv preprint arXiv:2501.14195.

**Ethical Concerns:**

["NO or VERY MINOR ethics concerns only"]

**Final Justification:**

Thanks for the detailed rebuttal. The additional experiments and explanations provided by the author have adequately addressed my main concerns about the submission. Therefore, I am inclined to accept it.

**Limitations:**

yes

**Quality:**

3

**Strengths And Weaknesses:**

Strengths:
1. This paper proposes a novel method that embeds graphical watermark into the video generation process, which diverges from methods that typically embed binary watermark. The novelty of adaptively embedding block-wise graphical watermark into generated video offers a meaningful contribution to the field of video watermarking.
2. The proposed framework is described in detail. The paper employs intuitive illustrations to clearly convey the motivation behind the adaptive matching mechanism, and the method section is well-structured and easy to understand.
3. Safe-Sora introduces the state space model into the field of video watermarking for the first time, along with a novel spatiotemporal local scanning strategy that effectively captures both intra-frame and inter-frame dependencies within video.

Weaknesses:
1. Although the paper claims to “embed graphical watermarks directly into the video generation process”, Fig. 2 and 7 indicate that the watermark embedding actually occurs during the transformation from latent representations to video (i.e., within the decoder). Therefore, I argue that categorizing this method as generative watermarking is somewhat misleading. In fact, the method adds the watermarking process to the decoder of latent diffusion model, which is conceptually closer to a post-processing watermarking method. The authors should more clearly articulate the methodological category to which their method belongs.
2. The proposed watermarking method requires training before deployment, yet the paper lacks a detailed comparison with baselines in terms of time or computational complexity—for instance, training time or the time required for watermark embedding.
3. In Section 4.1 on baseline selection, one of the baselines, PUSNet [1], is referred to by the authors as “a state-of-the-art image steganography network”. However, PUSNet is actually a model steganography method, designed to embed a steganography model into another cover model, rather than a method tailored for image steganography. Therefore, this comparison is inappropriate and potentially misleading.
4. In the robustness experiments presented in Section 4.3, the range of tested distortions is too limited. In practice, common attacks on video watermarking include frame insert, frame drop, frame swap, and random crop, none of which are evaluated in this paper. A more comprehensive robustness evaluation is necessary.

[1] Li, G., Li, S., Luo, Z., Qian, Z., & Zhang, X. (2024). Purified and unified steganographic network. In Proceedings of the IEEE/CVF conference on computer vision and pattern recognition (pp. 27569-27578).

---

> ### Author Rebuttal · Authors · 2025-07-29
>
> Thank you for your valuable comments and recognition on our novel method for embedding graphical watermarks into the video generation process, well-structured methodology with intuitive illustrations, and pioneering introduction of state space models to video watermarking. We hope the following discussion can address your concerns!
>
> ---
>
> > **Q1**: It is recommended to clearly elaborate on how the proposed method differs from post-processing watermarking methods.
>
> **A1**: Typical post-processing watermarking methods [1,2,3] inject watermarks into ready-made **visible images or videos.** For example, the classic post-processing method UDH [2] concatenates the watermark image and cover image, then inputs them into a UNet to produce a watermarked image.
>
> **However, Our method embeds watermarks during the VAE decoding process within video generation, when the video has not yet been exposed. This approach aligns with common practices in generative watermarking [4,5,6].**  For instance, the pioneering generative watermarking work Stable Signature [4] embeds watermarks in text-to-image generation by fine-tuning the VAE decoder. Similarly, WOUAF [5] injects watermark information into the VAE decoder through weight modulation techniques.
>
> Unlike previous generative image watermarking methods [4,5,6], for **generative video watermarking**, we propose *coarse-to-fine adaptive patch matching* to segment watermark images into patches and embed them into the most similar video frames. We also introduce a *spatiotemporal local scanning strategy* to enable enhanced fusion and extraction of watermark information across space and time.
>
> > **Q2**: A detailed analysis of the time or computational complexity should be provided.
>
> **A2**: We provide comprehensive computational cost analysis covering both training and testing phases, along with performance metrics to demonstrate cost-effectiveness.
> Training experiments are conducted on four RTX 4090 GPUs, while testing is performed on a single RTX 4090 GPU. Results are presented in the table below:
>
> |    | Balujanet | Wengnet | UDH     | HiNet   | PUSNet  | Safe-SD | Ours    |
> |-----------|-----------|---------|---------|---------|---------|---------|---------|
> | Training Time (h) | 8.06 | 8.89 | 7.4 | 23.93 | 29.47 | 43.37 | 38.16 |
> | Testing Time (s) | 0.0432 | 0.0494 | 0.2022 | 0.3441 | 1.1559 | 0.6817 | 0.3678 |
> | FVD↓ | 512.22 | 265.82 | 1075.62 | 125.74 | 154.35 | 849.83 | 3.77 |
>
> *Table 1: Computational speed between different methods.*
>
> **Testing Time**: During watermarked video generation, **the video generation process dominates the computational cost (46.8980s per video using VideoCrafter2 [7] on an RTX 4090), while watermark embedding time remains negligible across all methods.** Our method's testing time (0.3678s) is competitive with state-of-the-art approaches.
>
> **Training Time**: Although Safe-Sora requires the second-longest training time, it demonstrates strong cost-effectiveness when compared to competitive baselines. To properly assess this trade-off, it's important to note that FVD (Fréchet Video Distance) [8] serves as a key metric for evaluating watermarked video quality. The results reveal compelling evidence of efficiency: **compared to HiNet (the best-performing baseline with FVD 125.74), Safe-Sora requires only 1.6× training time while achieving dramatically superior performance—reducing FVD to just 3% of HiNet's value (3.77 vs. 125.74).** This substantial performance gain at modest computational overhead demonstrates a highly favorable performance-to-cost ratio, validating the practical efficiency of our approach.
>
> > **Q3**: It is advisable to include comparisons with methods specifically designed for image steganography, such as HiNet, and to replace unsuitable baselines PUSNet.
>
> **A3**: We appreciate this suggestion and have conducted comprehensive comparisons with HiNet, a state-of-the-art image steganography method. The detailed experimental results are presented below:
>
> | Method | PSNR ↑ | MAE ↓ | RMSE ↓ | SSIM ↑ | LPIPS ↓ |
> |-------|--------------------|-------------------|--------------------|--------------------|---------------------|
> | HiNet | 32.4               | 3.77              | 6.85               | 0.95               | 0.07                |
> | Ours  | **37.71**           | **2.22**          | **3.61**           | **0.97**           | **0.04**            |
>
> *Table 2: Watermark Quality Comparison with HiNet*
>
> | Method | PSNR ↑ | MAE ↓ | RMSE ↓ | SSIM ↑ | LPIPS ↓ | tLP ↓ | FVD ↓ |
> |--------|--------|--------|--------|--------|--------|--------|--------|
> | HiNet | 30.35 | 4.33 | 8.33 | 0.91 | 0.11 | 1.01 | 125.74 |
> | Ours     | **42.5** | **1.36** | **1.96** | **0.98** | **0.01** | **0.38** | **3.77** |
>
> *Table 3: Video Quality Comparison with HiNet.*
>
> The experimental results demonstrate that Safe-Sora significantly outperforms HiNet across all evaluation metrics. We will include HiNet in the final version to provide more comprehensive and appropriate comparisons.
>
> > **Q4**: In the robustness experiments, additional tests should be conducted under distortions such as random crop, frame insert, frame drop, and frame swap. Relevant experimental setups can refer to VideoShield.
>
> **A4**:  Due to time constraints, following VideoShields [9], we adopt temporal tamper (Frame Swap) and spatial tamper (Random Crop) to validate robustness, comparing our method against six baselines (including the additional HiNet baseline). We will include more comprehensive robustness experimental results in the final version. The preliminary results are shown below:
>
> | Method   | PSNR↑  | MAE↓  | RMSE↓ | SSIM↑ | LPIPS↓ |
> |----------|--------|-------|-------|-------|--------|
> | Balujanet| 25.33  | 9.56  | 15    | 0.91  | 0.11   |
> | Wengnet  | 33.37  | 3.64  | 5.68  | 0.96  | 0.06   |
> | UDH      | 22.85  | 11.37 | 19.33 | 0.74  | 0.28   |
> | HiNet    | 32.65  | 3.65  | 6.68  | 0.95  | 0.07   |
> | PUSNet   | 29     | 7.33  | 9.42  | 0.93  | 0.12   |
> | Safe-SD  | 25.11  | 8.74  | 15.8  | 0.86  | 0.09   |
> | Ours     | **37.53**  | **2.33**  | **3.69**  | **0.97**  | **0.05**   |
>
> *Table 4: Watermark Reconstruction Robustness under Frame Swap*
>
> | Method   | PSNR↑  | MAE↓  | RMSE↓ | SSIM↑ | LPIPS↓ |
> |----------|--------|-------|-------|-------|--------|
> | Balujanet| 17.46  | 21.9  | 35.58 | 0.62  | 0.31   |
> | Wengnet  | 13.9   | 38.2  | 53.72 | 0.45  | 0.48   |
> | UDH      | 6.53   | 101.2 | 121.58| 0.17  | 0.84   |
> | HiNet    | 9.84   | 75.54 | 84.48 | 0.4   | 0.58   |
> | PUSNet   | 14.98  | 33.55 | 47.39 | 0.49  | 0.45   |
> | Safe-SD  | 13.49  | 42.47 | 55.86 | 0.49  | 0.51   |
> | Ours     | **34.3**   | **3.15**  | **5.28**  | **0.96**  | **0.05**   |
>
> *Table 5: Watermark Reconstruction Robustness under Random Crop*
>
> The experimental results demonstrate that our method exhibits strong robustness, **outperforming all baselines across evaluation metrics**.
>
> > **Q5**: While the impact of Multi-Scale Feature Injection is explored, the strategy used for feature injection after removing this component is not mentioned. This should be clarified.
>
> **A5**:  In our implementation, the VAE decoder produces multi-scale features F1, F2, F3, and F4 at different resolution levels.
>
> **With Multi-Scale Feature Injection:** All multi-scale features (F1, F2, F3, and F4) are fused with the watermark feature maps, enabling comprehensive integration across different spatial scales.
>
> **Without Multi-Scale Feature Injection:** Only the final layer feature F4 is fused with the watermark feature maps, representing a single-scale injection strategy.
>
> We will polish this technical detail in the final version to ensure greater clarity.
>
> > **Q6**: In the right subfigure of Fig. 1, it is suggested to revise the y-axis label from "watermarking performance" to PSNR.
>
> **A6**: We will revise it in the final version to provide a clearer and more precise representation.
>
> > **Q7**: If a watermarked video is first encoded back into latent space using the same encoder, and then decoded again, would this process effectively remove the watermark?
>
> **A7**: We conducted robustness experiments to evaluate watermark preservation under VAE reconstruction (encode-decode process). The experimental results are presented below:
>
> | Method   | PSNR↑ | MAE↓  | RMSE↓ | SSIM↑ | LPIPS↓ |
> |----------|-------|-------|-------|-------|--------|
> | Balujanet | 16.04 | 24.57 | 42.01 | 0.57  | 0.44   |
> | Wengnet   | 7.27  | 103.1 | 112.76| 0.21  | 0.74   |
> | UDH       | 6.06  | 102.45| 127.65| 0.14  | 0.83   |
> | HiNet     | 9     | 79.98 | 93.21 | 0.33  | 0.65   |
> | PUSNet    | 8.81  | 76.9  | 95.53 | 0.26  | 0.71   |
> | Safe-SD   | 6.67  | 93.37 | 121.42| 0.39  | 0.84   |
> | Ours      | **28.32** | **6.42**  | **10.35** | **0.88**  | **0.15**   |
>
> *Table 6: Watermark Reconstruction Robustness under VAE Reconstruction*
>
> The results demonstrate that while all baseline methods suffer severe watermark degradation under this challenging attack, **Safe-Sora maintains superior watermark quality across all metrics**. In the future, we will explore the exclusive techniques to improve the watermarking robustness under this attack.
>
> ---
>
> [1] Hiding images within images. IEEE transactions on pattern analysis and machine intelligence 2019.
>
> [2] Udh: Universal deep hiding for steganography, watermarking, and light field messaging. NeurIPS 2020.
>
> [3] Purified and unified steganographic network. CVPR 2024.
>
> [4] The Stable Signature: Rooting Watermarks in Latent Diffusion Models. ICCV 2023.
>
> [5] WOUAF: Weight Modulation for User Attribution and Fingerprinting in Text-to-Image Diffusion Models. CVPR 2024.
>
> [6] LaWa: Using Latent Space for In-Generation Image Watermarking. ECCV 2024.
>
> [7] Videocrafter2: Overcoming data limitations for high-quality video diffusion models. CVPR 2024.
>
> [8] FVD: A new Metric for Video Generation. ICLR 2021.
>
> [9] VideoShield: Regulating Diffusion-based Video Generation Models via Watermarking. Arxiv 2025.

---

> > ### Comment · Reviewer_VQjW · 2025-08-05
> >
> > Thanks for the detailed rebuttal. The additional experiments and explanations provided by the author have adequately addressed my main concerns about the submission. Therefore, I am inclined to accept it.

---

> > > ### Author Response · Authors · 2025-08-06
> > >
> > > We are glad that our response has adequately addressed your main concerns about the submission. Thanks a lot for your comments and suggestions to improve our paper. We will incorporate them into the final version.

---

### Official Review · Reviewer_WgWy · 2025-06-26

**Clarity:** 2
**Significance:** 3
**Originality:** 3
**Rating:** 4
**Confidence:** 3

**Summary:**

This paper studied the solution to embed invisible graphic watermarks into the AI-generated videos. Specifically, this paper proposed a deep-learning-based method, which utilized and trained deep learning models to embed and extract the watermark. This paper proposed two new techniques. The first is a coarse-to-fine adaptive matching mechanism to select the most suitable frames to embed the watermark. The second is a Mamba-based block integrated into the deep learning models to extract the features of the video. Experimental results demonstrated that the proposed method could surpass existing methods.

**Questions:**

Please see the weaknesses.

**Ethical Concerns:**

["NO or VERY MINOR ethics concerns only"]

**Final Justification:**

The response addresses most of my concerns. I decided to maintain the positive rating.

**Limitations:**

Yes

**Paper Formatting Concerns:**

No.

**Quality:**

3

**Strengths And Weaknesses:**

Strengths:

1. The proposed two mechanisms are novel.
2. This paper designed an effective block based on Mamba.
3. The experimental results were promising.

Weaknesses:

1. The methodology section of this paper is confusingly written. While the authors focus heavily on their two proposed mechanisms, they lack a clear overall workflow description of the method. For instance, it's unclear how the watermarks are embedded and extracted. Furthermore, Section 3.4 on the loss function fails to specify which parameters are being optimized to minimize the objective. These omissions significantly reduce the readability of the paper.
2. The method proposed in this paper is a post-processing algorithm, meaning that the watermark embedding and extraction are independent of the video generation model. Therefore, in this context, I believe the paper lacks comparison or discussion with traditional video watermarking methods (those also independent of generation models).
3. Based on the results of the paper's ablation study, it appears that removing any single component has a minimal impact on the method's effectiveness. Both the watermark and video quality remain significantly higher than the baseline. The authors likely need to conduct further experiments to reveal which module truly contributes to the observed performance.
4. Potential overclaim: The paper's claim of being the first to embed image watermarks into AI-generated videos might be an overclaim, considering that bitstrings and images are interconvertible (via reshape and flatten operations).

---

> ### Author Rebuttal · Authors · 2025-07-29
>
> Thank you for your valuable comments and recognition on our novel two mechanisms, effective Mamba-based block design, and promising experimental results that surpass existing methods. We hope the following discussion can address your concerns!
>
> ---
>
> > **Q1**: The paper lacks a clear overall workflow description of the method. For instance, it's unclear how the watermarks are embedded and extracted. Furthermore, Section 3.4 on the loss function fails to specify which parameters are being optimized to minimize the objective.
>
> **A1**: The overall workflow of Safe-Sora is described in the caption of Figure 2. In general, our method consists of three stages: (1) **Watermark Preprocessing**: The Coarse-to-Fine Adaptive Patch Matching technique segments watermarks into patches, adaptively matches them to video locations, and upsamples them into feature maps; (2) **Watermark Embedding**: Watermark features integrate with video multi-scale features through a UNet with 2D SFMamba blocks, followed by 3D SFMamba blocks implementing our spatiotemporal scanning strategy; (3) **Watermark Extraction**: An extraction network with distortion layer, 3D SFMamba blocks, and position recovery module recovers the embedded watermark.
>
> **Optimization parameters**: All components are trainable except the frozen VAE encoder/decoder.
>
> We will polish the paper carefully in the final version to improve the clarity and readability.
>
> > **Q2**: The method proposed in this paper is a post-processing algorithm, meaning that the watermark embedding and extraction are independent of the video generation model. Therefore, in this context, I believe the paper lacks comparison or discussion with traditional video watermarking methods.
>
> **A2**:  Typical post-processing watermarking methods [1,2,3] inject watermarks into ready-made **visible images or videos.** For example, the classic post-processing method UDH [2] concatenates the watermark image and cover image, then inputs them into a UNet to produce a watermarked image. However, our method embeds watermarks **within the video generation process**, when the video has not yet been exposed. This approach aligns with common practices in generative watermarking [4,5,6].
>
> Unlike previous generative image watermarking methods [4,5,6], for **generative video watermarking**, we propose *coarse-to-fine adaptive patch matching* to segment watermark images into patches and embed them into the most similar video frames. We also introduce a *spatiotemporal local scanning strategy* to enable enhanced fusion and extraction of watermark information across space and time.
>
> In addition, **we compared our method with the traditional video watermarking method Wengnet, with results shown in Table 1 of the main paper.** Our experiments demonstrate that our approach comprehensively outperforms Wengnet across all metrics, particularly in temporal consistency (FVD [7]: 3.77 vs. 265.82)
>
> > **Q3**: The authors likely need to conduct further experiments to reveal which module truly contributes to the observed performance.
>
> **A3**: We have conducted ablation studies to evaluate the contribution of our two core components: Coarse-to-Fine Adaptive Patch Matching and Spatiotemporal Local Scanning.
>
> Coarse-to-Fine Adaptive Patch Matching: This component is motivated by our observation that watermark-image similarity significantly impacts watermarking performance. It segments the watermark image into patches and adaptively embeds them at different video locations in a coarse-to-fine manner. When this component is removed, both video quality and watermark recovery degrade substantially. For example, watermark quality deteriorates with RMSE increasing from 3.61 to 3.99, while video temporal consistency drops significantly with FVD increasing from 3.77 to 16.87.
>
> Spatiotemporal Local Scanning: This component leverages the frequency domain relationships in 3D wavelet transforms, performing scanning from low-to-high and high-to-low frequencies to enable enhanced fusion and extraction of watermark information across space and time. Without this component, temporal consistency is severely compromised, with FVD degrading from 3.77 to 13.16.
>
> These ablation results demonstrate that both components are essential for achieving optimal performance.
> We will include more comprehensive ablation experiments in the final version to provide a more thorough evaluation of each module's contribution.
>
> > **Q4**: The paper's claim of being the first to embed image watermarks into AI-generated videos might be an overclaim, considering that bitstrings and images are interconvertible (via reshape and flatten operations).
>
> **A4**: While bitstrings and images are interconvertible to some extent, **there is a fundamental difference in complexity and scale**. Previous works typically embed binary strings of 48 bits [4,6,8], whereas our method embeds 256×256 RGB images, which convert to 256×256×3×8 = 1,572,864 bits—approximately 32,000 times larger than traditional binary watermarks. **Therefore, embedding images can be viewed as an extremely high-capacity binary embedding task with significantly greater technical challenges.**
>
> Moreover, image watermarks offer distinct advantages over binary strings: they serve as more intuitive and visually recognizable evidence of ownership [2]. Such designs enhance both the perceptual clarity and practical reliability of copyright verification, making them more suitable for real-world applications.
>
> We acknowledge the reviewer's concern and will carefully revise this claim in the final version to more accurately reflect the technical contributions and significance of our work.
>
> ---
>
> [1] Hiding images within images. IEEE TPAMI 2019.
>
> [2] Udh: Universal deep hiding for steganography, watermarking, and light field messaging. NeurIPS 2020.
>
> [3] Purified and unified steganographic network. CVPR 2024.
>
> [4] The Stable Signature: Rooting Watermarks in Latent Diffusion Models. ICCV 2023.
>
> [5] WOUAF: Weight Modulation for User Attribution and Fingerprinting in Text-to-Image Diffusion Models. CVPR 2024.
>
> [6] LaWa: Using Latent Space for In-Generation Image Watermarking. ECCV 2024.
>
> [7] FVD: A new Metric for Video Generation. ICLR 2021.
>
> [8] LVMark: Robust Watermark for latent video diffusion models. Arxiv 2024.

---

> ### Comment · Reviewer_WgWy · 2025-08-07
>
> Thank you for the response, which addresses most of my concerns. I will maintain the positive rating.

---

> > ### Author Response · Authors · 2025-08-07
> >
> > Thank you for your positive feedback and for the constructive review process. We appreciate your valuable insights that have helped improve our work.

---

### Official Review · Reviewer_B5T6 · 2025-07-05

**Clarity:** 2
**Significance:** 2
**Originality:** 2
**Rating:** 4
**Confidence:** 2

**Summary:**

This paper introduces Safe-Sora, a new framework for embedding graphical watermarks directly into the video generation process to ensure copyright preservation. It leverages a hierarchical coarse-to-fine adaptive matching mechanism and pioneers a 3D wavelet transform-enhanced Mamba architecture to robustly embed and extract watermarks, achieving state-of-the-art performance in video quality, watermark fidelity, and robustness.

**Questions:**

I'm not an expert in this field. I will consider raise my score if the authors can address my conserns.

1. The proposed techniques, particularly the method of inserting watermarks by selecting specific video frames, appear to be directly borrowed from established image watermarking methods. The paper needs to clearly clarify which technical components are uniquely designed for video and justify their necessity beyond simple adaptation from the image domain. If the core techniques are indeed image-based, a comparison with state-of-the-art image watermarking solutions is need. Otherwise, the contribution that merely applying image techniques to video does not meet the bar for NeurIPS.

2. While Figure 4 shows an improved invisibility when differences are amplified, it also suggests that several existing methods already achieve near-imperceptible watermark invisibility in their original state. The paper needs to further justify the practical significance of its incremental improvements in invisibility if the baseline methods already satisfy human perception requirements.

3. The robustness evaluation primarily focuses on Gaussian blur and rotation. However, video compression is the most common processing applied to videos during transmission and storage. The authors should provide results about the robustness against various video compression standards to validate the practical utility of the method.

4. How about the computational overhead of the proposed method?

**Ethical Concerns:**

["NO or VERY MINOR ethics concerns only"]

**Final Justification:**

The authors have addressed most of my conserns.

**Limitations:**

Please see the questions.

**Quality:**

3

**Strengths And Weaknesses:**

Strengths:
1. This paper introduces a new method for embedding graphical watermarks directly into the video generation process.
2. The proposed method achieves superior video quality, watermark fidelity, and robustness against distortions.
Weaknesses:
1. The proposed techniques seem to be used in the image watermarking field.
2. The performance improvement is unclear. Prior methods have already achieved acceptable watermark and video quality.
3. Insufficient experimental results about the robustness of the watermark.

---

> ### Author Rebuttal · Authors · 2025-07-28
>
> Thank you for your valuable comments and recognition on our novel framework for embedding graphical watermarks directly into the video generation process and the superior performance in video quality, watermark fidelity, and robustness against distortions. We hope the following discussion can address your concerns!
>
> ---
>
> > **Q1**: The paper needs to clearly clarify which technical components are uniquely designed for video and justify their necessity beyond simple adaptation from the image domain. If the core techniques are indeed image-based, a comparison with state-of-the-art image watermarking solutions is need.
>
> **A1**: A static image does not contain any temporal information. Typical image watermarking methods input both the watermark image and the cover image into the watermark embedding network to generate a stego image [1, 2]. For example, the classic graphical watermarking method UDH [2] concatenates the watermark image and cover image, then inputs them into a UNet, producing a stego image. **Unlike images, videos consist of multiple frames, and Figure 1 shows that watermarking performance significantly correlates with the visual similarity between the watermark and cover images. Therefore, how to appropriately allocate the watermark across the various frames is a key challenge.**
>
> To address this challenge, Safe-Sora divides the watermark image into multiple patches and adaptively matches them to different positions across the frames. Furthermore, **considering the temporal relationships in videos**, we propose a spatiotemporal local scanning strategy to enhance the fusion and extraction of watermark information across both space and time.
>
> Additionally, we compare Safe-Sora with state-of-the-art image watermarking solutions, such as PUSNet [3] and HiNet [4]. The comparison with PUSNet is provided in Table 1 of the main paper, and the comparison with HiNet is discussed in the following tables:
>
> | Method | PSNR ↑ | MAE ↓ | RMSE ↓ | SSIM ↑ | LPIPS ↓ |
> |-------|--------------------|-------------------|--------------------|--------------------|---------------------|
> | HiNet | 32.4               | 3.77              | 6.85               | 0.95               | 0.07                |
> | Ours  | **37.71**           | **2.22**          | **3.61**           | **0.97**           | **0.04**            |
>
> *Table 1: Watermark Quality Comparison with HiNet*
>
>
> | Method | PSNR ↑ | MAE ↓ | RMSE ↓ | SSIM ↑ | LPIPS ↓ | tLP ↓ | FVD ↓ |
> |--------|--------|--------|--------|--------|--------|--------|--------|
> | HiNet | 30.35 | 4.33 | 8.33 | 0.91 | 0.11 | 1.01 | 125.74 |
> | Ours     | **42.5** | **1.36** | **1.96** | **0.98** | **0.01** | **0.38** | **3.77** |
>
>
> *Table 2: Video Quality Comparison with HiNet.*
>
>
> Experimental results show that our method outperforms these state-of-the-art image watermarking approaches, highlighting the necessity of our video-tailored architectural innovations.
>
>
> > **Q2**: The paper needs to further justify the practical significance of its incremental improvements in invisibility if the baseline methods already satisfy human perception requirements.
>
> **A2**:  Figure 4 presents single-frame image comparison results. However, **videos fundamentally differ from single-frame images in their temporal dynamics.** Even when watermarks appear nearly imperceptible in individual frames, they can introduce temporal inconsistencies across multiple video frames, significantly degrading overall video quality and failing to meet human perceptual standards.
>
> **Qualitative Analysis:** Due to space constraints, Figure 5 visualizes multi-frame qualitative results exclusively for Safe-Sora. We will include comprehensive multi-frame qualitative comparisons with all baseline methods in the final version to provide a more complete evaluation.
>
> **Quantitative Analysis:** We employ FVD (Fréchet Video Distance) [5] and tLP [6] metrics to quantitatively assess video temporal consistency. As shown in Table 1 of the main paper, Safe-Sora substantially outperforms all baselines across both metrics. Notably, Safe-Sora achieves an FVD of 3.77, dramatically surpassing baseline methods (e.g., PUSNet: 154.35, Wengnet: 265.82).
>
> These results clearly demonstrate that existing baseline methods exhibit significant limitations in maintaining temporal consistency, while **Safe-Sora effectively preserves temporal coherence in watermarked videos—a critical requirement for practical video watermarking applications.**
>
> > **Q3**: The authors should provide results about the robustness against various video compression standards to validate the practical utility of the method.
>
> **A3**: We follow the experimental setup for video compression robustness from previous literature [7, 8] and use the widely used H.264 compression to validate the robustness of our method, with the results shown in Figure 6. The results show that under H.264 compression, all baseline methods suffer a significant drop in performance, whereas **our method maintains high watermark quality**. In the final version of the paper, we will conduct more video compression robustness experiments to validate the practical utility of our method.
>
> > **Q4**: How about the computational overhead of the proposed method?
>
> **A4**: We provide comprehensive computational cost analysis covering both training and testing phases, along with performance metrics to demonstrate cost-effectiveness.
> Training experiments are conducted on four RTX 4090 GPUs, while testing is performed on a single RTX 4090 GPU. Results are presented in the table below:
>
> |    | Balujanet | Wengnet | UDH     | HiNet   | PUSNet  | Safe-SD | Ours    |
> |-----------|-----------|---------|---------|---------|---------|---------|---------|
> | Training Time (h) | 8.06 | 8.89 | 7.4 | 23.93 | 29.47 | 43.37 | 38.16 |
> | Testing Time (s) | 0.0432 | 0.0494 | 0.2022 | 0.3441 | 1.1559 | 0.6817 | 0.3678 |
> | FVD↓ | 512.22 | 265.82 | 1075.62 | 125.74 | 154.35 | 849.83 | 3.77 |
>
> *Table 3: Computational speed between different methods.*
>
> **Testing Time**: During watermarked video generation, **the video generation process dominates the computational cost (46.8980s per video using VideoCrafter2 [9] on an RTX 4090), while watermark embedding time remains negligible across all methods.** Our method's testing time (0.3678s) is competitive with state-of-the-art approaches.
>
> **Training Time**: Although Safe-Sora requires longer training time, it demonstrates strong cost-effectiveness when compared to competitive baselines. To properly assess this trade-off, it's important to note that FVD (Fréchet Video Distance) serves as a key metric for evaluating watermarked video quality. The results reveal compelling evidence of efficiency: **compared to HiNet (the best-performing baseline with FVD 125.74), Safe-Sora requires only 1.6× training time while achieving dramatically superior performance—reducing FVD to just 3% of HiNet's value (3.77 vs. 125.74).** This substantial performance gain at modest computational overhead demonstrates a highly favorable performance-to-cost ratio, validating the practical efficiency of our approach.
>
> ---
>
> [1] Hiding images within images. IEEE transactions on pattern analysis and machine intelligence 2019.
>
> [2] Udh: Universal deep hiding for steganography, watermarking, and light field messaging. NeurIPS 2020.
>
> [3] Purified and unified steganographic network. CVPR 2024.
>
> [4] Hinet: Deep image hiding by invertible network. ICCV 2021.
>
> [5] FVD: A new Metric for Video Generation. ICLR 2021.
>
> [6] Learning temporal coherence via self-supervision for gan-based video generation. ACM TOG 2020.
>
> [7] ItoV: Efficiently Adapting Deep Learning-based Image Watermarking to Video Watermarking. International Conference on Culture-Oriented Science and Technology 2023.
>
> [8] LVMark: Robust Watermark for latent video diffusion models. Arxiv 2024.
>
> [9] Videocrafter2: Overcoming data limitations for high-quality video diffusion models. CVPR 2024.

---

> > ### Author Response · Authors · 2025-08-06
> >
> > Dear Reviewer,
> >
> > Thank you for your comprehensive feedback and for raising important questions regarding the practical significance of our improvements, computational overhead, and other concerns.
> >
> > We hope our rebuttal has sufficiently addressed the points you raised. If any aspects require further explanation or if additional concerns have emerged, please let us know and we will be happy to provide clarification.
> >
> > We appreciate your time and look forward to your response.
> >
> > Best regards,
> >
> > Authors

---

### Official Review · Reviewer_xHnf · 2025-07-05

**Clarity:** 3
**Significance:** 3
**Originality:** 3
**Rating:** 4
**Confidence:** 1

**Summary:**

The paper introduces a framework for embedding graphical watermarks into the video generation process of text-to-video diffusion models, called safe-sora. The authors propose a coarse-to-fine adaptive matching strategy that embeds watermark patches into visually similar spatiotemporal locations in the video.

**Questions:**

* How resilient is Safe-Sora to adversarial attempts to detect or remove the embedded watermark? Have you tested watermark removal attacks (e.g., inpainting, adversarial noise, filtering)?
* Could Safe-Sora be extended to embed dynamic watermarks (e.g., animated logos or video clips) instead of static images? (Even if not implemented, a discussion of the technical challenges and design considerations for dynamic watermarks would be valuable.)

**Ethical Concerns:**

["NO or VERY MINOR ethics concerns only"]

**Final Justification:**

I appreciate the authors' feedback. My concerns are mostly addressed, hence I will keep my score as I believe it still adequately reflects my evaluation of the paper.

**Limitations:**

Yes

**Quality:**

3

**Strengths And Weaknesses:**

The problem of watermarking video-generation is important and relevant, it addresses, among others, the question of copyright protection for generated video content. To the best of my knowledge (which is limited in the domain), this seems to be the first application of state space models (Mamba) and 3D wavelet transforms for video watermarking, combined with a sophisticated patch matching mechanism. The evaluation and analysis, as well as the ablations, seem comprehensive; the authors use benchmarks against image watermarking, video steganography, and generative watermarking baselines, showing clear improvements in watermark recovery and video quality, as well as demonstrating resilience to various real-world distortions, including compression, blur, and erasing.

There is currently no discussion as to whether a knowledgeable adversary could detect or remove the watermark, covering this question seems important to discuss the method in full. Further, the method is currently restricted to static image watermarks; dynamic or more complex watermarks (e.g., videos) are left for future work, and embedding performance relies on matching the watermark to semantically similar video regions, which may limit generalisability.

---

> ### Author Rebuttal · Authors · 2025-07-31
>
> Thank you for your valuable comments and recognition on our method novelty combining Mamba and 3D wavelet transforms, comprehensive evaluation and ablations, and clear improvements in watermark recovery and robustness. We hope the following discussion can address your concerns!
>
> ---
>
> > **Q1**: Have you tested watermark removal attacks？
>
> **A1**: We conducted robustness experiments on watermarking under various attacks (H.264 compression, Gaussian noise, blur, etc.), with the results presented in Figure 6.  **Our experimental findings consistently demonstrate that Safe-Sora significantly outperforms all baseline methods, showcasing superior robustness capabilities.** Notably, under H.264 compression—a critical real-world scenario—all baseline methods experience substantial performance degradation, while our method maintains consistently high watermark extraction quality, validating its practical applicability for robust copyright protection.
>
> > **Q2**: Could Safe-Sora be extended to embed dynamic watermarks (e.g., animated logos or video clips) instead of static images?
>
> **A2**: Safe-Sora's design principles naturally support extension to dynamic watermarks. Our method leverages content similarity analysis to segment watermark images into patches and adaptively embed them across optimal video positions. Dynamic watermarks (animated logos or video clips) can be conceptualized as multi-frame image sequences. **Our adaptive patch matching technique can similarly decompose these multi-frame sequences into spatial-temporal patches and intelligently select the most suitable embedding positions based on content similarity analysis.** This represents a promising and technically feasible direction that we plan to explore as a key focus in our future research endeavors.

---

> > ### Author Response · Authors · 2025-08-06
> >
> > Dear Reviewer,
> >
> > Thank you for your detailed review and for the thoughtful questions about watermark removal attacks and the potential for dynamic watermark embedding.
> >
> > We would greatly appreciate it if you could let us know whether our responses have adequately addressed your concerns, or if there are any remaining issues that need further clarification.
> >
> > Thank you for your time and consideration.
> >
> > Best regards,
> >
> > Authors

---

> > > ### Comment · Reviewer_xHnf · 2025-08-06
> > >
> > > I appreciate the authors' feedback. My concerns are mostly addressed, hence I will keep my score as I believe it still adequately reflects my evaluation of the paper.

---

### Comment · Area_Chair_c2n6 · 2025-08-05
**Reminder: Discussion and Final Justification**

Dear Reviewers,

As we approach the end of the author–reviewer discussion phase (Aug 6, 11:59pm AoE), I kindly remind you to read the author rebuttal carefully, especially any parts that address your specific comments.

Please consider whether the response resolves your concerns, and if not, feel free to engage in further discussion with the authors while the window is still open.

Your timely participation is important to ensure a fair and constructive review process. If you feel your concerns have been sufficiently addressed, you may also submit your Final Justification and update your rating early. Thank you for your contributions.

Best,

AC

---

### Note · Authors · 2025-08-14

Dear AC,

We sincerely appreciate all your efforts in coordinating the review of our submission.

In our paper, we introduce Safe-Sora, the first framework to embed graphical watermarks directly into the video generation process for reliable copyright preservation of AI-generated content. Safe-Sora demonstrates significant potential to advance video watermarking research and protect intellectual property in the era of generative AI.

As the reviewers highlighted, our work is:

**Well-motivated, fulfilling a clear practical significance** for copyright protection in AI-generated video content, addressing important and relevant questions of copyright protection for generated video content. (Reviewer `xHnf`, `VQjW`)

**Technically novel, offering innovative approaches** including the first application of state space models (Mamba) and 3D wavelet transforms for video watermarking, with novel mechanisms and meaningful contributions to the field. (Reviewer `xHnf`, `WgWy`, `VQjW`)

**Supported by comprehensive experiments**, demonstrating superior performance in video quality, watermark fidelity, and robustness against various real-world distortions. Our method significantly outperforms baseline methods across evaluation metrics. (Reviewer `xHnf`, `B5T6`, `VQjW`)

We are thankful for the constructive feedback from reviewers and have carefully replied to all comments. Specifically, we have:

- **Clarified technical distinctions** by explaining how our generative watermarking approach differs from post-processing methods and providing detailed workflow descriptions.

- **Provided comprehensive performance analysis**, including detailed computational cost analysis, comparisons with additional state-of-the-art baselines (HiNet), and extensive robustness evaluations against various attacks including frame swap, random crop, and VAE reconstruction.

- **Addressed scope and limitations** by discussing extension possibilities to dynamic watermarks, clarifying the scale difference between our image watermark embedding (1.5M+ bits) versus traditional binary watermarks (48 bits), and acknowledging areas for future improvement.

We will revise the final manuscript accordingly to improve clarity and better highlight our technical contributions. We sincerely hope these updates will better deliver the benefits of the proposed Safe-Sora framework.

Thank you for your consideration, and we look forward to your decision.

Best regards,
Authors

---

### Decision · Program_Chairs · 2025-09-17

**Decision:**

Accept (poster)

**Comment:**

This work proposed Safe-Sora, the first framework to embed graphical watermarks directly into the video generation process for reliable copyright preservation of AI-generated content. Safe-Sora demonstrates significant potential to advance video watermarking research and protect intellectual property in the era of generative AI. The authors propose a coarse-to-fine adaptive matching strategy that embeds watermark patches into visually similar spatiotemporal locations in the video, pioneering a 3D wavelet transform-enhanced Mamba architecture to robustly embed and extract watermarks.

This work has two significant strengths as follows.

- Novelty. This paper proposes a novel method that embeds graphical watermark into the video generation process, which diverges from methods that typically embed binary watermark. The novelty of adaptively embedding block-wise graphical watermark into generated video offers a meaningful contribution to the field of video watermarking.
- Comprehensive empirical analysis. The evaluation and analysis, as well as the ablations, seem comprehensive; the authors use benchmarks against image watermarking, video steganography, and generative watermarking baselines, showing clear improvements in watermark recovery and video quality, as well as demonstrating resilience to various real-world distortions, including compression, blur, and erasing.

The reviewers mentioned multiple concerns including adversary detection, limited scope, and technical clarity. The authors have addressed most concerns raised by the reviewers to some extent. This is a complete and novel work.

Given the consensus that three reviewers gave the borderline accept, I recommend Accept (Poster).